# Phase synchronization of fluid-fluid interfaces as hydrodynamically coupled oscillators

Eujin Um [1✉], Minjun Kim [1], Hyoungsoo Kim [2], Joo H. Kang [3], Howard A. Stone [4] & Joonwoo Jeong [1✉]

Hydrodynamic interactions play a role in synchronized motions of coupled oscillators in fluids, and understanding the mechanism will facilitate development of applications in fluid mechanics. For example, synchronization phenomenon in two-phase flow will benefit the design of future microfluidic devices, allowing spatiotemporal control of microdroplet generation without additional integration of control elements. In this work, utilizing a characteristic oscillation of adjacent interfaces between two immiscible fluids in a microfluidic platform, we discover that the system can act as a coupled oscillator, notably showing spontaneous in-phase synchronization of droplet breakup. With this observation of in-phase synchronization, the coupled droplet generator exhibits a complete set of modes of coupled oscillators, including out-of-phase synchronization and nonsynchronous modes. We present a theoretical model to elucidate how a negative feedback mechanism, tied to the distance between the interfaces, induces the in-phase synchronization. We also identify the criterion for the transition from in-phase to out-of-phase oscillations.

[1] Department of Physics, Ulsan National Institute of Science and Technology (UNIST), Ulsan 44919, Republic of Korea. [2] Department of Mechanical Engineering, Korea Advanced Institute of Science and Technology (KAIST), Daejeon 34141, Republic of Korea. [3] Department of Biomedical Engineering, Ulsan National Institute of Science and Technology (UNIST), Ulsan 44919, Republic of Korea. [4] Department of Mechanical and Aerospace Engineering, Princeton University, Princeton, NJ 08544, USA. ✉email: eujinum@unist.ac.kr; jjeong@unist.ac.kr

Synchronization in nature is universal, as the interaction of oscillators can occur in various forms and across a wide range of length and timescales[1–9]. Although commonly observed, understanding the mechanism of the synchronization still raises challenging questions, often requiring to solve non-linear differential equations, even for the phase synchronization of two identical oscillators[10]. The common examples of identical oscillators found in a living room, such as pendulum clocks[11], and multiple metronomes ticking at the same frequency[12], exhibit phase synchronization if two or more bodies are coupled through appropriate mediators. Many examples of synchronization found in nature involve complex biological or physiological coupling factors[3,13–16], and it is often impossible to identify all of the elements that are associated with synchronization. Designing diverse experimental model systems with adjustable parameters in isolation from each other should deepen our understanding of synchronization[17–20].

Synchronization is also observed in the interaction of swimming organisms[21–27]. For example, the emergence of in-phase or anti-phase synchronization and collective behaviors has been studied theoretically and experimentally, with a controversy over whether the dominant effect arises from a hydrodynamic origin or biochemical pathways[14,22,28–32]. Therefore, a model physical system to study hydrodynamically coupled synchronization at low Reynolds numbers, where viscous forces are dominant over inertial forces, can help separate the role of hydrodynamic interactions from other (e.g., biochemical) complexity[33]. Micro-fluidic systems characterized by low Reynolds numbers allow the isolated study of interfacial tension, elastic, and viscous forces, with no interference from turbulence, by controlling parameters such as flow rates, viscoelasticity of the fluids, and the dimension or geometry of the channel structures. For example, the theoretical model of waving infinite sheets, first explored by G. I. Taylor, lays the stepping stones for understanding flagellar synchronization of cells swimming in proximity to each other, in terms of minimum energy dissipation[26]. Other examples of experimental models of hydrodynamic coupling include colloidal oscillators in an optical trap[18], elastic cylinders in a viscoelastic flow[34], rotating helices[35], chiral propellers[17], and oscillating bubbles with a constant heat source[36]. Developing an experimental model to achieve phase locking and investigate a transition between different modes of synchronization in waves along fluid–fluid interfaces will enhance our understanding of the role of hydrodynamic interactions for the synchronization behaviors in fluids.

Microdroplets (or bubbles) generated from oscillating interfaces in a microchannel can serve as periodic oscillators useful for the study of hydrodynamic synchronization[37–39]. The mechanism of droplet generation in a T-channel of confined geometry has been studied frequently[40–45]. The periodic droplet generation in a microchannel has enabled the development of various micro-fluidic applications, including droplet manipulation[39,46], droplet pairing[47–49], and oscillation of droplets in groups[50]. In a micro-fluidic configuration where two interfaces interact[51], including the so-called double-T-junction geometry[52,53], the implementation of one-by-one generation of droplets has triggered various applications that exploit the controlled delivery of droplets of two different compositions[53–57]. This mode of one-by-one generation can be mapped into the out-of-phase synchronization mode of droplet breakup from each branch, and the convergence of the droplet generation frequency between the two branches having slightly different flow rates has been investigated, although the specific times of the droplet breakup do not coincide[58]. We note that all of the previous research on microfluidic droplet generation with two interacting interfaces only revealed the presence of this "out-of-phase" generation of droplets.

In this work, we complete the state diagram of droplet generation in a double T junction by discovering the uncharted regime of in-phase synchronization of droplet breakup from two nearby multiphase interfaces, as well as a nonsynchronous regime. We also develop a theoretical model to explain how the symmetric coincidence of droplet-breakup times from the two branches is stabilized in the in-phase synchronization regime. We investigate experimentally each regime according to the strength of the hydrodynamic coupling by varying the distance between the interfaces and the flow rate of each phase. Our theoretical model illustrates that the forces exerted on the two different interfaces are coupled by the size of the gap through which the continuous phase flows, and the coupling eventually leads to spontaneous in-phase or out-of-phase synchronization of droplet generation, depending on the capillary number that describes the flow. This droplet-based system can serve as a model of a coupled oscillator encompassing all of the main dynamical regimes[10], covering in-phase and out-of-phase synchronization, and non-synchronous regimes, based on the effect of hydrodynamic interactions between the interfaces.

## Results

**Emergence of synchronization modes from two interfaces.** Two fluid–fluid interfaces at a double T junction of a microfluidic configuration exhibit various flow regimes according to the flow conditions. As shown in Fig. 1a, when the dispersed phases (water) from two branches facing each other enter the main channel filled with the continuous phase (oil), two interfaces form at the double T junction; see Supplementary Fig. 1 for a schematic diagram of the microfluidic device. The flow rate of the oil is $Q_o$, and the flow rate of the water in each branch is $Q_w$, which is divided equally from a water inlet owing to the symmetric design.

When there is no interaction between the two interfaces, each branch generates droplets independently. With the width of the main channel $w_c$ as wide as 1500 μm (height $h = 40$ μm) under the constant flow rate, we observe the decoupled droplet generation from two branches (Supplementary Fig. 2a), with two distinct frequency peaks in the Fourier analysis of the droplet-breakup time series (Supplementary Fig. 2b); we conduct the statistical analysis with more than 500 droplets per each flow-rate condition. The sharpness of the distinct peaks indicates that each branch generates droplets consistently but independently with no interference or phase locking between them. The difference in drop-generation frequencies between the two branches is on average 7.27 ± 3.71% from 18 different flow-rate conditions, indicating that the same $Q_w$ does not guarantee the symmetric droplet generation mode. The coefficient of variance of the droplet generation frequency, i.e., the standard deviation divided by the average, is 13.5% with a syringe pump[59], and 6.3% with a pressure pump.

Smaller $w_c$ brings the two interfaces closer and we start to see the interaction between them. The relative flow rate of the dispersed and continuous phases $Q_w Q_o^{-1}$ plays a crucial role in the determination of the interacting flow regime. As we lower $Q_w Q_o^{-1}$, from the jetting regime of sufficiently high $Q_w Q_o^{-1}$[52,53], the interfaces begin to oscillate and generate droplets periodically in a one-by-one sequence from each branch (Fig. 1a, b, and Supplementary Movie 1). The Fourier analysis shows that each branch exhibits a sharp frequency peak, which exactly overlaps to each other (Supplementary Fig. 3a), and we designate this regime the out-of-phase state. If we keep decreasing $Q_w$, creating droplets of smaller volume, we observe a transition regime where the droplet breakup from the two branches is occasionally in-phase, i.e., the two branches each generate a droplet at the same time. (Fig. 1c, d, and Supplementary Movie 2). In this transition regime

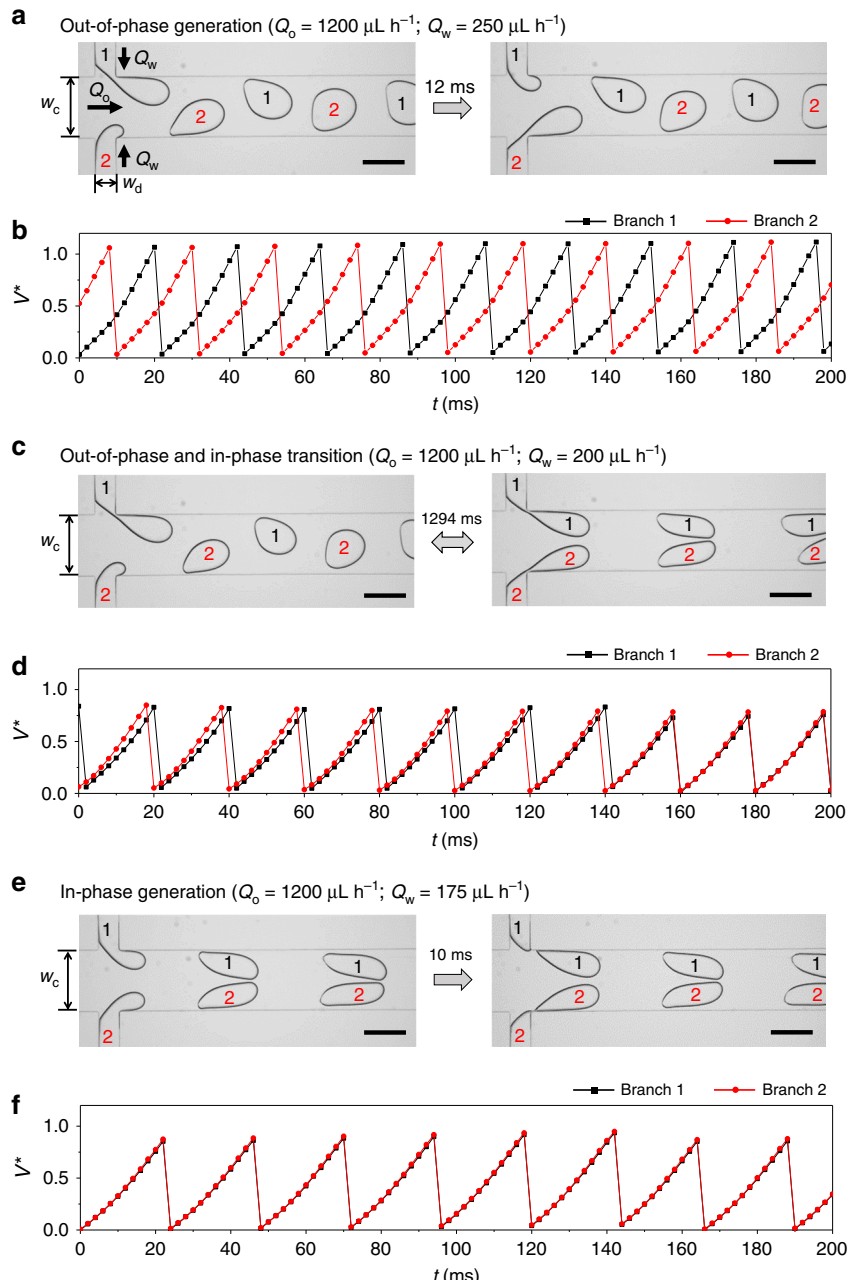

**Fig. 1 The different modes of droplet breakup at the double T junction.** Upon decreasing the flow rate of the dispersed phase, $Q_w$, the droplet generation mode changes from the (**a**, **b**) out-of-phase synchronization mode to the (**c**, **d**) transition mode, and then to the (**e**, **f**) in-phase synchronization mode, under the condition of a fixed continuous-phase flow rate, $Q_o$. Snapshots of movies recorded from a microscope are shown with a given time interval (**a**, **c**, **e**). The plots show how the scaled volume $V^*$ of the dispersed phase from each branch changes according to the time (**b**, **d**, **f**); the volume is scaled by the volume of the junction $w_c w_d h$, where $w_c$, $w_d$, $h$ are the widths of the main channel and branch, and the height of the channel, respectively. The time-varying $V^*$ from two branches exactly overlap in (**f**). All scale bars indicate 200 μm.

between the out-of-phase and in-phase regimes (O–I transition), the centers of the frequency peaks overlap, demonstrating the existence of coupling between interfaces, but the peaks broaden because the state oscillates between the two synchronized modes with different frequencies despite the constant $Q_w$ and $Q_o$ (Supplementary Fig. 3b).

The in-phase synchronization state is stabilized by further reducing $Q_w$, and in this regime the droplet-breakup time is always identical for the two branches; such hydrodynamically coupled interfaces in this configuration are first reported in this

work (Fig. 1e, f, and Supplementary Movie 3). As in the out-of-phase synchronization mode, each branch in the in-phase mode exhibits a sharp frequency peak in the power spectrum, which overlaps exactly with the other within ±0.01% both in the in-phase and out-of-phase synchronization regimes due to phase locking (Supplementary Fig. 3c). Further reduction of $Q_w$ will scramble this in-phase droplet generation and a state of a nonsynchronous regime appears.

Both of in-phase and out-of-phase synchronization states are robust to the initial condition or temporary perturbations of the

flow. For example, if we disrupt the flow by pressing down the channel or the tubing, and release the disturbance, the synchronized state is recovered within a few seconds (Supplementary Movie 4). The droplets generated in-phase can flow side-by-side along the main channel as long as 15 mm, which is 50 times the channel width ($w_c = 300$ µm) (Supplementary Fig. 4a, b and Supplementary Movie 5), although the configuration can be disrupted downstream at lower $Q_w$ or higher $Q_o$ conditions, with smaller droplets and larger differences in velocities (Supplementary Fig. 4c).

We study the phase locking of two oscillating interfaces in the synchronization regime by analyzing the time series of droplet generation. With $T_1$ and $T_2$, the periods of droplet breakup from branch 1 and branch 2, being the same in the in-phase and out-of-phase synchronization states (Supplementary Fig. 3a, c), we define a synchronization parameter $\alpha$,

$$\alpha = \frac{\max(\Delta t_{12}, T_1 - \Delta t_{12})}{T_1},\tag{1}$$

which is the normalized phase difference between the two branches (Supplementary Fig. 5). $\Delta t_{12}$ is the time delay between the breakup from branches 1 and 2. In the case of two identical coupled oscillators, ideally only two modes are allowed: in-phase ($\alpha = 1$) and anti-phase ($\alpha = 0.5$) synchronization[10]. However, in our experimental system, variations in $\alpha$ exist. From the values of $\alpha$ obtained from the experiments, we categorize $\alpha$ into two states, as the in-phase ($0.8 < \alpha \le 1$) and out-of-phase synchronization modes ($0.5 \le \alpha \le 0.8$) (Supplementary Fig. 6). The in-phase and out-of-phase states are robust to temporary perturbations of the flow, as mentioned above; hence the phase difference, $\alpha$, is also recovered after the transient perturbed state, manifesting the existence of coupling between the interfaces. This phase locking between two oscillators, regardless of initial conditions, is the hallmark of phase synchronization[10]. $\alpha$ is not defined in the nonsynchronous and O–I transition regimes since the order of the droplet-breakup sequence in the two branches keeps changing during the observation time of these regimes, thus making $\Delta t_{12}$ meaningless.

**Synchronization depends on the distance between interfaces.** We develop state diagrams of droplet breakup according to flow conditions in various channel widths, $w_c = 100$, 300, and 400 µm (Fig. 2a–c). The different $w_c$ results in a different distance between the interfaces, hence affecting the coupling strength. The state diagram of the interface is shown as a function of $Q_w Q_o^{-1}$ and the capillary number, $Ca = \frac{\mu_o v}{\gamma}$, where $\mu_o$ is the dynamic viscosity of the continuous oil phase, $\gamma$ is the interfacial tension between oil and water phases, and the average flow velocity $v$ is estimated as $Q_o$ divided by the area $hw_c$ of the channel cross section. The state of each data point is determined by analyzing time-series data of the minimum 500 droplets generated from the T junction.

All the state diagrams (Fig. 2a–c) exhibit the same trend regardless of the channel width $w_c$: the droplet generation mode changes from the jetting, out-of-phase synchronization modes to the in-phase mode as $Q_w Q_o^{-1}$ is lowered. At much lower $Q_w Q_o^{-1}$, a nonsynchronous regime appears. Controlling the flow rates with a pressure pump instead of the syringe pumps does not affect the trend (Supplementary Fig. 7). As $w_c$ increases, the in-phase synchronization regime expands over a broader range of Ca, while the out-of-phase regime shrinks and almost disappears when $w_c = 400$ µm. Above $w_c > 400$ µm, the nonsynchronous behaviors are manifested (Supplementary Fig. 2). This strong dependence on the distance between the interfaces hints at the vital role of hydrodynamic coupling for the synchronization of droplet generation.

We discover a strong correlation between the synchronization parameter $\alpha$ (Eq. (1)) and the maximum protrusion height of the dispersed phase in the main channel. We measure the protrusion height $b(t)$ at time $t$ to determine the actual distance between the interfaces, and the maximum height $b_{max}$, right before the droplet breakup (Fig. 2d). Figure 2e illustrates the relationship between the normalized parameter $b_{max}^* = b_{max} w_c^{-1}$ and $\alpha$, in all experimental conditions. When $b_{max}^* \ge 0.5$, the out-of-phase synchronization mode occurs, and the in-phase synchronized mode occurs when $b_{max}^* \le 0.4$. When $0.4 < b_{max}^* < 0.5$, the in-phase and out-of-phase mode co-exists. The mode of O–I transition also occurs at this range of $b_{max}^*$, although $\alpha$ cannot be defined in the transition regime. This result suggests that the coupling between the interfaces in terms of separation distance is weaker during the in-phase synchronization mode compared to the out-of-phase mode. Decoupling of interfaces, i.e., the nonsynchronous regime, is observed for $b_{max}^* < 0.2$. When the channel is as wide as $w_c = 1500$ µm, $b_{max}^* < 0.2$ at all flow-rate conditions, which results in nonsynchronous behaviors only.

**Modeling the in-phase synchronization of the droplet breakup.** To explain the reason why the in-phase synchronization of droplet breakup is dominantly observed below a critical value of $b_{max}^*$ and how two interfaces are coupled, we develop a force-balance model of droplet breakup at the interface[41,43]. As shown in Fig. 3a, the process of droplet formation consists of filling and necking stages. The dispersed phases protrude in a symmetric manner from two opposing branches into the main channel of width $w_c$. During the filling stage, the dispersed phase flows from the branch into the main channel to the point defined by the parameter $b_{fill}$, and as the necking stage starts, the continuous phase squeezes the dispersed phase, collapsing the neck until a droplet breaks off. We determined the beginning of the necking stage by observing the change in the moving direction of the oil–water interface contacting the left wall of the branches (blue arrows, Fig. 3a). There is also a lag stage before the filling stage, in which the dispersed phase recedes, just after droplet breakup[43]; however, we included this stage within our definition of filling stage because its duration was negligible under most of our experimental conditions. The relationship between $\alpha$ and $b_{fill}^* = b_{fill} w_c^{-1}$ (Supplementary Fig. 8) is similar to that between $\alpha$ and $b_{max}^*$ (Fig. 2e). The transition from the in-phase to the out-of-phase synchronization states begins at about $b_{fill}^* = 0.2$.

During the filling stage, the continuous phase flows through the gap of width $w_g$, between the two evolving interfaces. When $b$ is the protrusion height of the dispersed phase, $\mu_o$ is the dynamic viscosity of the continuous phase, and $h$ is the height of the channel, the forces acting on the protruded dispersed phases include shear forces, $F_\tau \approx \tau bh \approx \frac{\mu_o Q_o b}{w_g h}$, where the shear stress is $\tau \approx \frac{\mu_o Q_o}{w_g h^2}$. The force arising from the increased resistance to flow of the continuous phase, $F_R \approx \Delta P bh \approx \frac{12 \mu_o Q_o b^2}{w_g^2 h}$, also exists where $\Delta P$ is the pressure drop across the droplet approximated from a lubrication analysis[42,43]. We adopted these approximate results for forces from a single-T-junction configuration and applied them to the double-T-junction design with the gap width $w_g$. By contrast, the capillary force $F_\gamma \approx -\gamma h$, resulting from the interfacial tension $\gamma$ between the continuous and dispersed phases, resists the breakup of the droplet. Therefore, as $w_g$ narrows during the filling stage, the breakup forces exerted on the dispersed phase, $F_\tau$ and $F_R$, increase and eventually equal the resisting interfacial tension force, $F_\gamma$, i.e., $F_\tau + F_R + F_\gamma = 0$. This moment marks the end of the filling stage, and determines

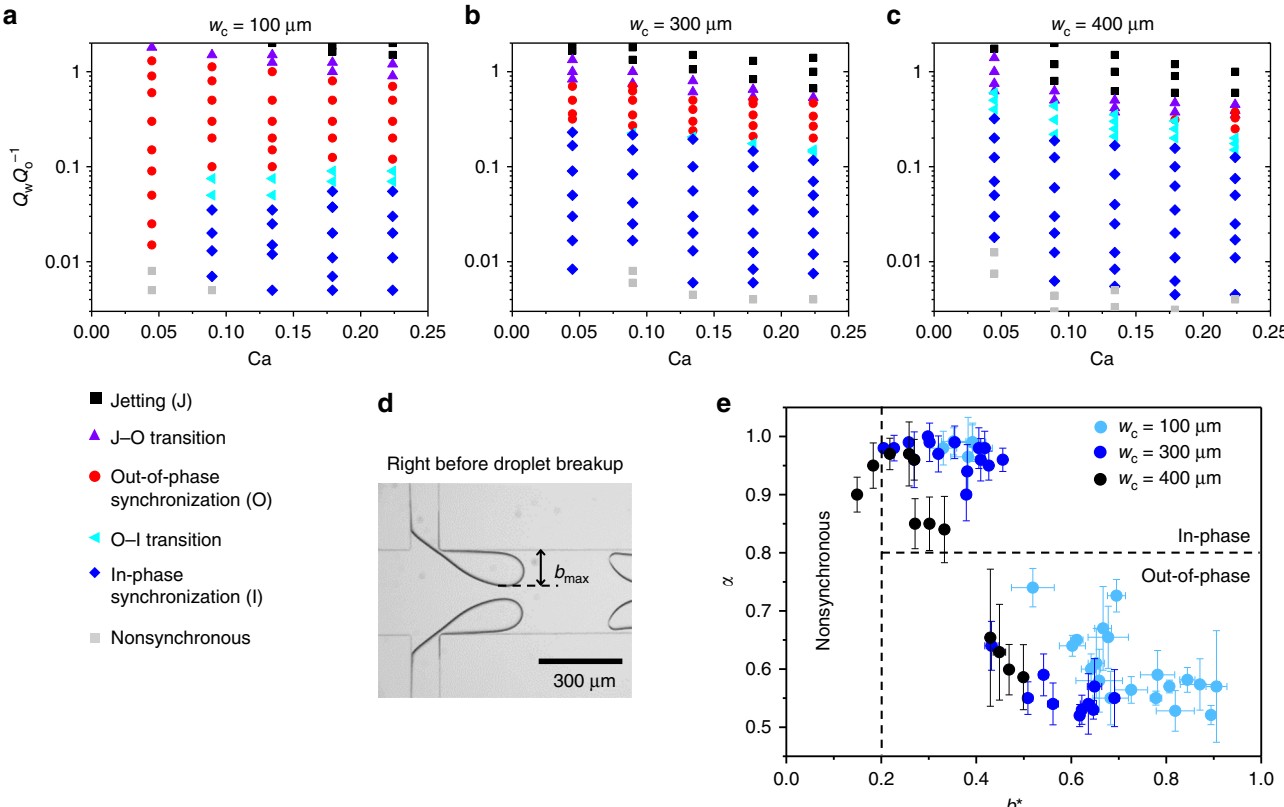

**Fig. 2 Study of parameters determining the in-phase or out-of-phase synchronization. a–c** State diagrams showing the relationship between the modes of synchronization, capillary number Ca, and the ratio of the flow rates $Q_w Q_o^{-1}$ according to different main-channel widths, $w_c$. The out-of-phase synchronization regime (red disks) is dominant when (**a**) $w_c = 100\,\mu m$, and the in-phase synchronization regime (blue diamonds) becomes dominant as the channel widens, for example, **b** $w_c = 300\,\mu m$, and **c** $w_c = 400\,\mu m$. The regimes of jetting (black squares), the transition between jetting and out-of-phase mode (purple triangles; J–O transition), the transition between the out-of-phase and in-phase synchronization mode (light blue triangles; O–I transition), and nonsynchronous states (gray squares) are also indicated. **d** A snapshot showing the maximum protrusion height $b_{max}$ of the water phase at the end of the necking stage, right before the droplet breaks. **e** A plot of the synchronization parameter, $\alpha$ (defined as Eq. (1) in the main text), versus $b_{max}^* = b_{max} w_c^{-1}$. Each data points represent the average values of $\alpha$ and $b_{max}^*$ measured from time-series data of the minimum 500 droplets; the error bars indicate standard deviations. The horizontal dashed line indicates the criterion, $\alpha = 0.8$, which discriminates the in-phase and out-of-phase synchronization states. Nonsynchronous states occur below $b_{max}^* = 0.2$, as shown with the vertical dashed line.

$b = b_{fill}$. During the following necking stage, $b$ continues to increase, while the continuous phase squeezes and elongates the neck of the dispersed phase for the duration $T_{neck}$. At the end of the necking stage, the droplet breaks off, leaving the branch.

The protrusion height of one dispersed phase affects the breakup force on the other dispersed phase by changing $w_g$, through which the continuous phase flows. Therefore, the gap width, $w_g$, between the two interfaces is the hydrodynamic coupling factor that induces and stabilizes the synchronized breakup of the dispersed phases. In our model, we calculated the protrusion height $b_i(t)$ as a function of time $t$ for each branch, $b_1(t)$ for branch 1 and $b_2(t)$ for branch 2. The gap width is defined as $w_g = w_c - [b_1(t) + b_2(t)]$. For a given initial condition, i.e., $b_1(0)$ and $b_2(0)$, and a given flow rate of the continuous phase ($Q_o$), we can solve for the times, $t_{fill,1}$ and $t_{fill,2}$, at which the net force acting on the interface becomes zero in each phase. For the flow rate of the dispersed phase, we adopt the average $\frac{db_i}{dt}$ from experimental data as a constant for simplicity, assuming that the protrusion height increases linearly and $\frac{db_1}{dt} = \frac{db_2}{dt}$. Representative experimentally measured $b_i(t)$ is shown in Fig. 3b, with more examples of $b_i(t)$ in channels of different $w_c$ provided in Supplementary Fig. 9. After the necking stage with a given duration $T_{neck}$, $b_i(t)$ is reset to 0, and the calculation is repeated to give $b_i(t)$ for the subsequent

cycles. Normally, the larger $Q_o$ expedites the necking by increasing the force on the dangling dispersed phase and shortens $T_{neck}$[41,43]; our experimental results also exhibit the same trend (Supplementary Fig. 10). However, a full understanding of $T_{neck}$ is not available, and we adopt the values of $T_{neck}$ from the experimental measurements.

A representative result of the calculation with $w_c = 300\,\mu m$ is shown in Fig. 3c. For this calculation, the experimental parameters from the data in Fig. 3b are adopted by the model. Our model demonstrates that the values of $b_i(t)$ for the two interfaces converge to give the in-phase synchronization state within a few cycles, regardless of the initial conditions; the ends of the filling stages ($t_{fill,1}$ and $t_{fill,2}$) are marked as dashed lines in Fig. 3c. The following analytical argument with Fig. 3d (expanded view of Fig. 3c) shows that the key mechanism leading to the in-phase synchronization is the negative feedback via the gap width $w_g$. If $b_2(t) = b_1(t) + (\Delta b)_n$, during the filling stage of the $n$th cycle, the dispersed phase from branch 2 enters the necking stage earlier than that of branch 1 by the time difference $(\Delta t_{fill})_n$. In this case, $b_{1,fill}$ cannot exceed $b_{2,fill}$ because the gap width $w_g$ will continue to decrease during $(\Delta t_{fill})_n$, resulting in larger force from the continuous phase acting on $b_1(t)$ during the filling stage. Since $\frac{db}{dt}$ and $T_{neck}$ for both of the dispersed phases are the same, $b_{1,fill}$

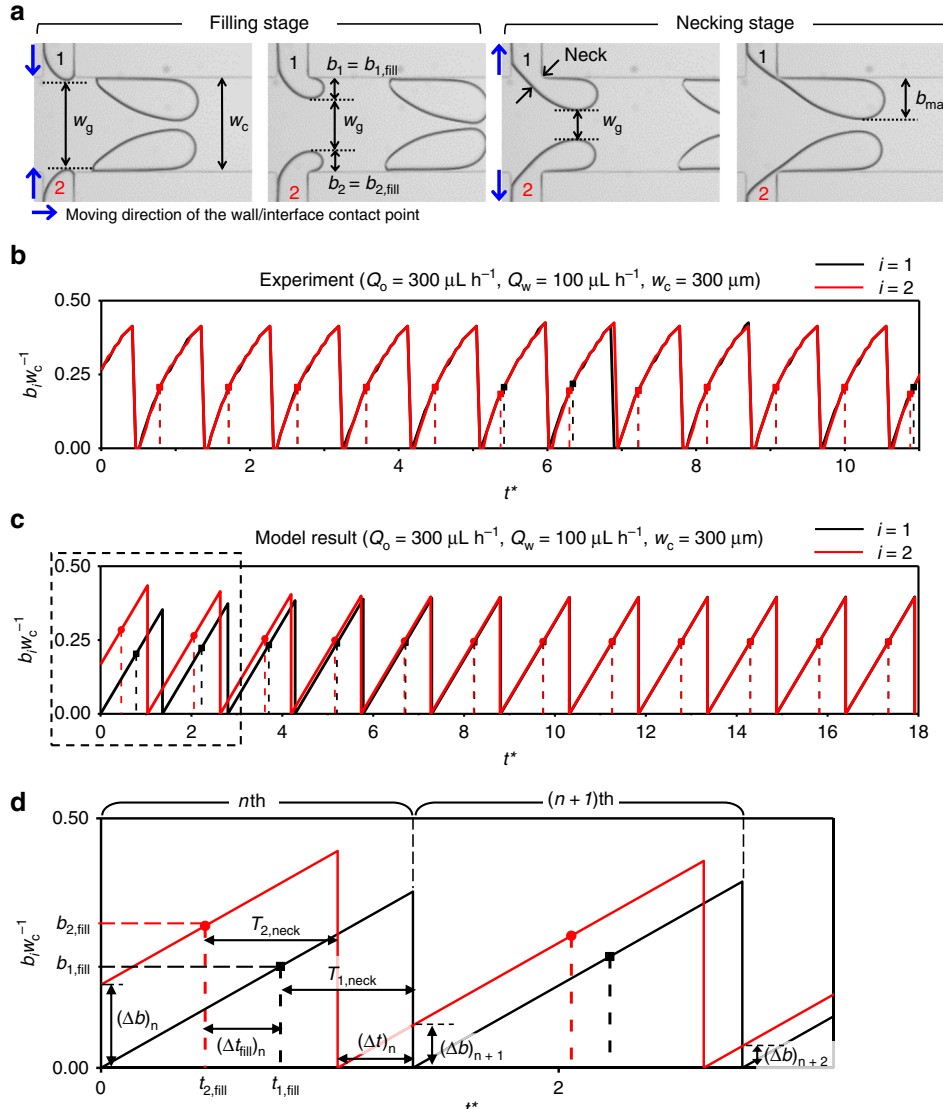

**Fig. 3 Development of the in-phase synchronization model from experimental parameters. a** Geometry of droplet breakup during the in-phase synchronization mode. The protrusion height into the main channel of width $w_c$ is shown as $b_i$ where $i$ is the branch index, and the size of the gap between the interfaces is $w_g$. The blue arrows indicate the moving direction of the contact point of the interface with the branch wall during the filling stage and the necking stage. At the end of the necking stage, the protrusion height reaches $b_{max}$ and the droplet leaves the branch. **b** Experimentally measured protrusion height (normalized), $bw_c^{-1}$, as a function of dimensionless time ($t^* = t\frac{Q_w}{w_c \cdot w_d h}$). The dots on the curves indicate the end of *the* filling stage $t_{fill}$ in each cycle and the corresponding $b_{fill}$. The data from the two branches almost overlap. **c** Model calculation showing the stability of the in-phase synchronization mode. The evolution of the protrusion heights shows how the initially nonsynchronous state converges to the state of perfect in-phase synchronization. **d** Expanded view of the data shown in the dashed box in (**c**), showing the variables to explain the stabilizing mechanism of the in-phase synchronization; the definition of each variable is written in the main text. Each cycle is defined by one droplet-breakup sequence from branch 1, as indicated by brackets on top of the graph. The difference in $b$ between two branches, $(\Delta b)_n$ where $n$ denotes the order of the cycle, decreases with each cycle, eventually leading to the in-phase synchronization.

and $b_{2,fill}$ can be expressed as

$$b_{2,fill} = (\Delta b)_n + \frac{db}{dt} t_{2,fill}, \qquad (2)$$

$$b_{1,fill} = \frac{db}{dt} t_{1,fill} \qquad (3)$$

In addition, the difference in protrusion during the next ($n + 1$)th cycle is

$$(\Delta b)_{n+1} = (\Delta t)_n \cdot \frac{db}{dt} = (\Delta t_{fill})_n \cdot \frac{db}{dt}, \qquad (4)$$

since $(\Delta t_{fill})_n + T_{1,neck} = T_{2,neck} + (\Delta t)_n$, and $T_{1,neck} = T_{2,neck}$. By

subtracting Eq. (3) from (2), and applying Eq. (4), we obtain the following Eq. (5),

$$\begin{aligned} b_{2,fill} - b_{1,fill} &= (\Delta b)_n + \frac{db}{dt} t_{2,fill} - \frac{db}{dt} t_{1,fill} \\ &= (\Delta b)_n - \frac{db}{dt}(\Delta t_{fill})_n = (\Delta b)_n - (\Delta b)_{n+1}. \end{aligned} \qquad (5)$$

Because $b_{2,fill} > b_{1,fill}$, we can conclude $(\Delta b)_n > (\Delta b)_{n+1}$.

Since $(\Delta b)_{n+1}$ is always smaller than $(\Delta b)_n$, the difference in the protrusion heights at the start of each cycle decreases toward zero until the $b_1(t)$ and $b_2(t)$ completely synchronize in-phase. The negative feedback affecting the breakup forces are linked through the gap width so as to stabilize the in-phase synchronization state

by suppressing the difference in protrusion heights between the two interfaces, $\Delta b$. We explore our model using various experimental conditions, including the flow rate of the continuous phase $Q_o$, the rate of change of the protrusion height $\frac{db}{dt}$, and the channel width $w_c$, and we confirm that the in-phase synchronization is robust as long as the gap width $w_g = w_c - (b_1(t) + b_2(t)) > 0$, which plays a pivotal role in determining the force balance on the interfaces. We note that this convergence to the in-phase synchronized state is insensitive to the detailed forms of $F_\tau$ and $F_R$ adopted from the literature[43]. The convergence occurs as long as the breakup force increases with a decrease in the coupled gap width $w_g$, satisfying the criterion that $b_{1,\text{fill}}$ cannot exceed $b_{2,\text{fill}}$ when $b_2(t) > b_1(t)$.

**Transition from the in-phase to the out-of-phase state**. Our experimental results reveal the emergence of in-phase synchronization modes of droplet breakup from two nearby interfaces, and our theoretical model of the coupled interfaces supports the stability of this in-phase synchronization. As shown in the state diagrams plotted as a function of Ca and $Q_w Q_o^{-1}$ (Fig. 2a–c), the synchronized state exhibits a transition to the out-of-phase state mainly according to the flow ratio $Q_w Q_o^{-1}$, with additional dependence on $w_c$. The model of droplet breakup introduced above to illustrate the stability of the in-phase synchronization mode can also explain this transition semiquantitatively. The transition boundary between the in-phase and out-of-phase modes can be derived by finding the relation between Ca and $Q_w Q_o^{-1}$ corresponding to $b_{\max}^* = \frac{db}{dt} \cdot (T_{\text{fill}} + T_{\text{neck}}) w_c^{-1} = 0.5$. We chose $b_{\max}^* = 0.5$ ideally, because above $0.5 w_c$, it is obvious that the droplets from the two interfaces cannot break up at the same time, although the experimental data shows that the transition boundary lies broadly between $0.4 < b_{\max}^* \leq 0.5$ (Fig. 2e). We rearrange the equation into

$$\frac{db}{dt} \cdot T_{\text{neck}} = 0.5 w_c - \frac{db}{dt} \cdot T_{\text{fill}}, \tag{6}$$

where $T_{\text{fill}}$ in the synchronized mode can be calculated from the force-balance equation when the net force acting on the droplet becomes zero, with $b_2(t) = b_1(t)$, and a constant $\frac{db_1}{dt}$. For the left-hand side of Eq. (6), we can approximate $\frac{db}{dt} \propto Q_w(w_d + w_c)^{-1}h^{-1}$, because the protrusion rate of the dispersed phase is related to the flow rate $Q_w$ and the channel dimensions, i.e., the channel width $w_c$ and branch width $w_d$ (Supplementary Fig. 11). $T_{\text{neck}}$ is proportional to $w_d w_c h Q_o^{-1}$, since the neck of the dispersed phase shrinks faster as the continuous phase flows faster and $T_{\text{neck}}$ decreases as the branch width $w_d$ gets narrower (Supplementary Fig. 10)[41,43]. Therefore, $\frac{db}{dt} \cdot T_{\text{neck}}$ on the left-hand side of Eq. (6) is proportional to $w_d w_c (w_d + w_c)^{-1} \cdot Q_w Q_o^{-1}$. After replacing $T_{\text{fill}}$ with the solution of the force-balance equation, the equation of the transition boundary is written in terms of $\text{Ca} = \frac{\mu_o Q_o}{\gamma w_c h}$ as

$$\frac{Q_w}{Q_o} \cdot \frac{1}{1 + \Lambda} = A \cdot \frac{6}{11 + \sqrt{1 + \frac{48h}{w_c \text{Ca}}}}, \tag{7}$$

where $\Lambda = w_c w_d^{-1}$ is the width ratio, and $A$ is a coefficient of proportionality. The detailed derivation of the right-hand side with the solution of $T_{\text{fill}}$ is in Supplementary Note 1. In our experiments, $0.05 < \text{Ca} < 0.25$, then the effective flow-rate ratio $\frac{Q_w}{Q_o} \cdot \frac{1}{1+\Lambda}$ where the transition occurs is approximately proportional to $\sqrt{\text{Ca} \frac{w_c}{h}}$.

We plot the experimental data from Fig. 2a–c with the effective dimensionless numbers, i.e., $\text{Ca} \cdot w_c h^{-1}$ and $Q_w Q_o^{-1}(1 + \Lambda)^{-1}$, and compare it with the trend of Eq. (7) (Fig. 4), choosing

$A = 0.12$ to best describe the collapsed data. Our experimental observations of the transition between the out-of-phase and the in-phase synchronization regimes (O–I transition) qualitatively agree with the transition line corresponding to $b_{\max}^* = 0.5$ predicted from the model. Because the actual O–I transition occurs between $0.4 < b_{\max}^* \leq 0.5$, the transition boundary in the experimental state diagrams is rather broad. Moreover, in this prediction of the transition boundary, we assume $T_{\text{neck}}$ is inversely proportional to $Q_o$, since there is no accurate theory available to predict $T_{\text{neck}}$. However, as observed from experiments, the variation in $T_{\text{neck}}$ gets larger as $Q_o$ is reduced, which may result in a wider transition regime towards lower Ca. We also expect that different dynamics of droplet breakup may be involved for larger $w_c$. As seen from the difference in the limit of $b_{\max}^*$ in each confinement, e.g., $w_c = 400\,\mu\text{m}$ and $1500\,\mu\text{m}$, $b_{\max}^*$ never exceeds 0.5 and 0.2, respectively, regardless of flow rates, which cannot be explained with the current model. Since $b_{\max}^*$ is an important parameter to determine the transition from the alternating to synchronized states, the mechanism of determining $b_{\max}^*$ in each confinement should be investigated further.

**Approximate modeling of the out-of-phase state**. The above-mentioned force-balance model only explains the stability of the in-phase synchronization mode of droplet breakup. To explain the stability of the out-of-phase mode, we propose a modified droplet-breakup model. If both of the dispersed phases protrude up to the point where $w_g$ is close to zero, but the breakup force from the continuous phase is not strong enough to break the droplet, the two interfaces cannot maintain the in-phase synchronized state, hence undergo a transition to the out-of-phase state. We can explain the out-of-phase model in a semi-quantitative manner by modifying the effective gap width. As shown in Fig. 5a, the symmetry of the breakup geometry in the two interfaces is broken, thus the gap width between them cannot be defined as in the in-phase mode. As illustrated in Fig. 5a, if $b_2(t) > b_1(t)$, the narrowest gap for dispersed phase 1 in the filling stage is affected by the gap between dispersed phase 2 and the channel wall, $w_c - b_2$, and is comparable in size. We assumed that the gap for dispersed phase 1 may be expressed as $w_{g,1} = w_c - \xi \cdot b_2(t)$, with a correction factor $\xi$. To mimic the experimental data of the out-of-phase state shown in Fig. 5b ($\alpha = 0.53$), we selected $\xi = 1.05$ empirically. After $b_2(t)$ is reset to 0, we set $w_{g,1} = w_c - b_{2,\max}$ until the end of the filling stage of $b_1(t)$. The duration of the necking stage, $T_{\text{neck}}$, is adopted from the experimental measurements. Applying this approximation to the model, we found that $b_1(t)$ and $b_2(t)$ converge to the out-of-phase synchronization mode, having a fixed value of $\alpha$, which depends on $\xi$ (Fig. 5c). Further work on ideas toward an improved model to determine the synchronization parameter $\alpha$ is given in the Supplementary Note 2.

**Investigating the phase synchronization with flow asymmetry**. Our first observation of the in-phase synchronization of droplet breakup from two interfaces in a microchannel occurs with the same source of flow rates as to create nearly identical oscillators, with the intrinsic variance of ~10% in the droplet generation frequency from a T-channel[59]. We investigate the effect of flow asymmetry on the appearance of in-phase synchronization by applying different pressures ($P_1$ on branch 1 and $P_2$ on branch 2) independently to each branch in $w_c = 300$ and $400$-$\mu$m channels (Fig. 6a). The pressure pump is used in these experiments for a relatively lower variation in the flow rates compared to the syringe pumps. After setting $P_1 = P_2$ where the in-phase

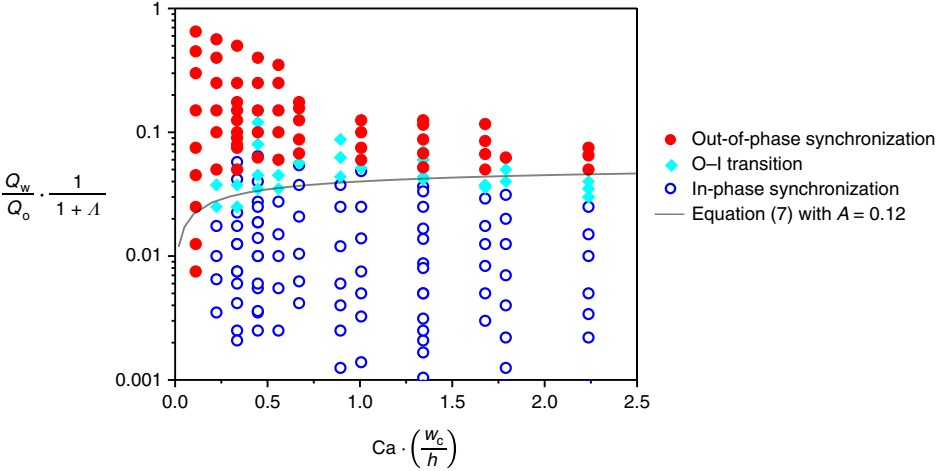

**Fig. 4 State diagram of the synchronization mode with effective dimensionless numbers.** All experiment results from different widths ($w_c$) of the main channel were combined and plotted in relation to effective Capillary number, $\mathrm{Ca} \cdot \frac{w_c}{h}$ and effective flow-rate ratio, $\frac{Q_w}{Q_o} \cdot \frac{1}{1+\Lambda}$, where $h$ is the channel height, and $\Lambda = w_c w_d^{-1}$ ($w_d$, the width of branch channel). The transition line predicted from Eq. (7) shows where $b_{max}^* = 0.5$, and the coefficient of proportionality $A = 0.12$ is chosen to best describe the transition in the state diagram.

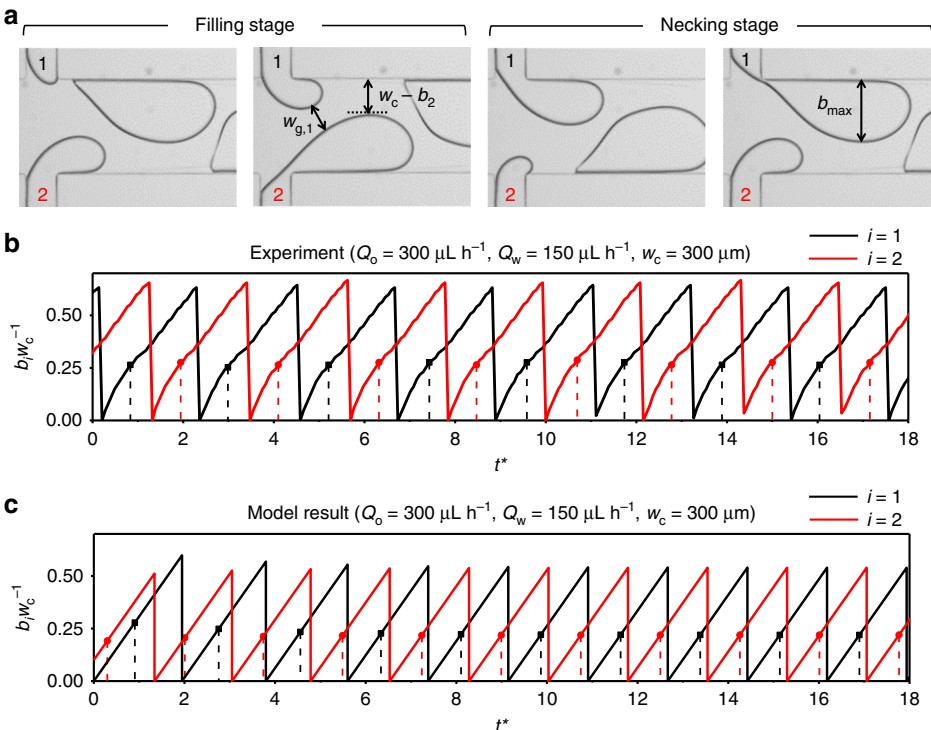

**Fig. 5 Modeling of the out-of-phase synchronization mode. a** Droplet-breakup geometry of the out-of-phase mode. We define the smallest gap width between the interfaces as $w_{g,1}$, which affects the filling stage of the dispersed phase 1. The distance between the dispersed phase 2 and the wall of the main channel is $w_c - b_2$. Considering the similarity between these two lengths, we assume $w_{g,1} = w_c - \xi \cdot b_2$ with a correction factor $\xi$. **b** Experimentally measured protrusion heights (normalized), $bw_c^{-1}$, as a function of dimensionless time ($t^* = t \cdot \frac{Q_w}{w_c w_d h}$). The dots on the curves indicate the end of the filling stage in each cycle and the corresponding $b_{fill}$. **c** Model calculation showing the stability of the out-of-phase state. The evolution of the protrusion heights shows how the initial state, arbitrarily chosen, converges to the out-of-phase state of a constant synchronization parameter $\alpha$. The experimental parameters from the data in (**b**) are adopted for the model calculation.

synchronization occurs (Fig. 6b) at every fixed pressure $P_o$ to the oil phase, we only vary $P_2$. The in-phase synchronization persists while the volume difference between two droplets reaches up to 75% (Fig. 6c–e). While satisfying the criterion for the in-phase synchronization, i.e., $0.8 < \alpha \le 1$, the value of $\alpha$ tends to be larger when $P_2 > P_1$ from the branch 2, compared to $P_2 < P_1$ (Fig. 6e). We presume that the production of larger droplets with higher $P_2$,

gives more time to stabilize the in-phase synchronization between interfaces compared to the smaller droplets. The dependence of $\alpha$ on the volumetric configuration of droplets hints that the deviations in $\alpha$ from the ideal value $\alpha = 1$ in in-phase synchronization shown in Fig. 2e may stem from a slight symmetry breaking despite the symmetric design sharing the same flow source. As the asymmetry in the branches increases with

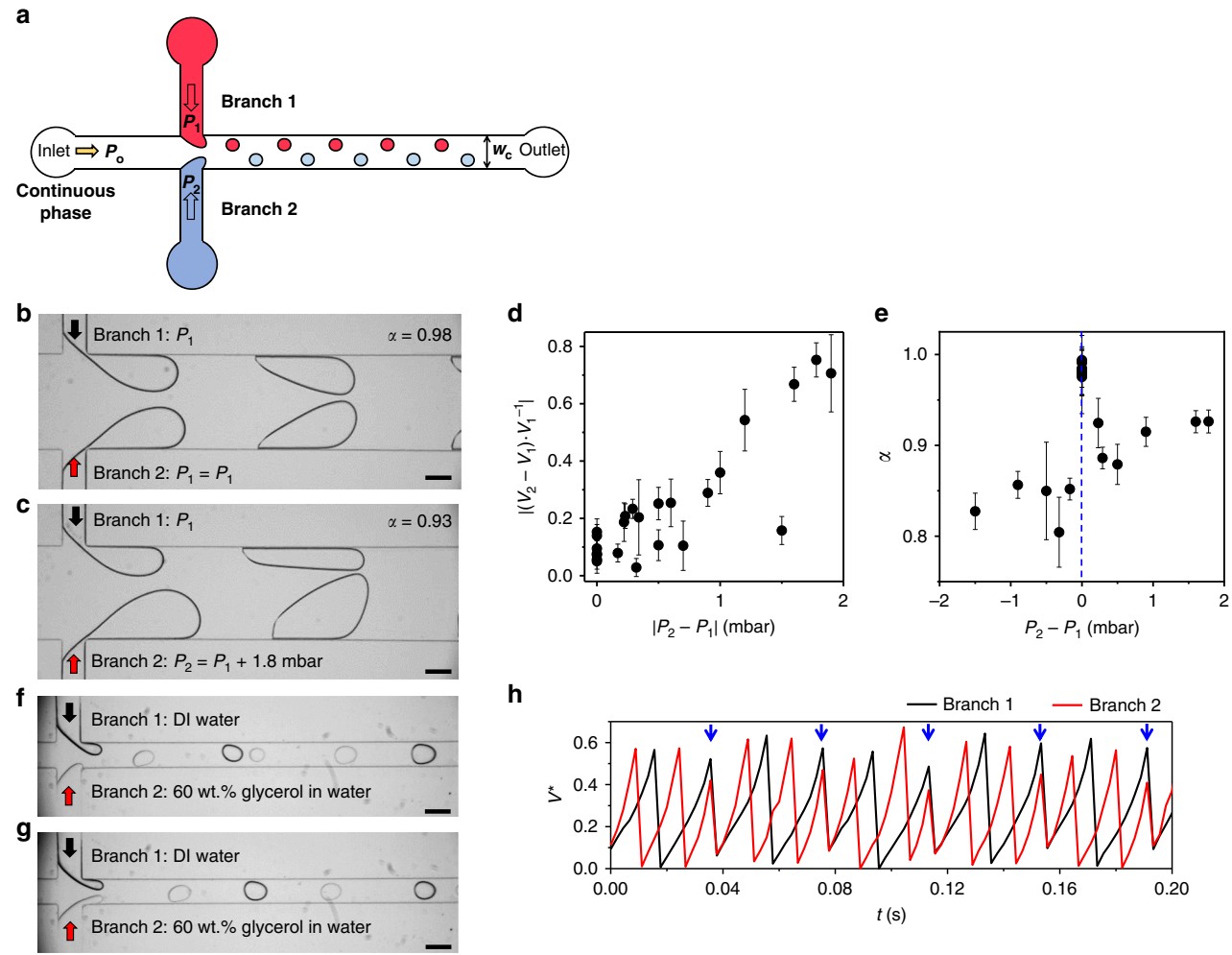

**Fig. 6 Droplet generation with two independently controlled dispersed phases. a** Schematic diagram showing the two dispersed phases controlled independently in opposing branches of the double T junction. **b–e** In-phase synchronization of droplet generation through the independently controlled branches while the pressure applied to branch 1 ($P_1$) is fixed, and only $P_2$ is varied. Representative microscope images ($P_O = 341$ mbar and $P_1 = 301$ mbar) are shown when (**b**) $P_1 = P_2$ with $\alpha = 0.98$, and (**c**) when $P_1 \neq P_2$ (bottom), which maintained the in-phase synchronization state with $\alpha = 0.93$ despite the considerable difference in the sizes of droplets. **d** The difference in the volume of droplets from two branches ($V_1$ from branch 1 and $V_2$ from branch 2) and **e** the synchronization parameter $\alpha$, is shown in relation to the difference between $P_1$ and $P_2$. For (**d**, **e**), each data points represent the average values of $\alpha$ and $b^*_{max}$ measured from time-series data of the minimum 200 droplets; the error bars indicate standard deviations. **f–h** An example of droplet generation with different viscosities ($Q_o = 500$ μL h$^{-1}$, and $Q_{w,1} = Q_{w,2} = 20$ μL h$^{-1}$). After three droplets are generated in the (**f**) out-of-phase mode, (**g**) two droplets are produced in the in-phase synchronization mode. **h** This 3:2 ratio of droplet generation from two branches is shown with the evolution of normalized volumes of each dispersed phase as a function of time. Blue arrows indicate the moment when the droplet breakups from the two interfaces coincide.

$|P_2 - P_1| > 2$ mbar, the in-phase synchronization is perturbed, and eventually the O–I transition and the out-of-phase regime dominate.

For more practical use of our system with various fluids, we further assign different viscosities for the two dispersed phases (Fig. 6f–h). In this system, the out-of-phase mode of synchronization prevails, and when the difference between the two viscosities becomes as large as a factor of 10, the dispersed phase with the higher viscosity produces more droplets as shown in Fig. 6h, where a 3:2 droplet generation ratio is observed. We find that this *m:n* droplet-breakup mode, in fact, consists of both the out-of-phase and in-phase modes (indicated with blue arrows in Fig. 6h), manifesting the existence of hydrodynamic coupling between the interfaces. Understanding this phenomenon requires a model beyond our force-balance approach, which assumes symmetric dispersed phases. Additional effects from the difference in the viscosity or the flow rates of the dispersed phase need further investigation.

## Discussion

Under the influence of strong coupling strength tied to the gap width ($b^*_{max} \geq 0.5$), the out-of-phase synchronization mode tolerates larger variations in the intrinsic frequencies than the in-phase mode, as similar to the previously reported work on droplet generation in parallel systems with slightly different flow rates[58]. In order to achieve the in-phase synchronization, the coupling strength, which depends on the distance between the interfaces, has to be decreased ($b^*_{max} < 0.5$) and the frequencies of the two oscillating interfaces have to be similar, in the range of small variations. Our findings resemble a series of observations in hydrodynamically coupled oscillators[18,23]. For instance, two flagella on two separate cells dominantly display the out-of-phase synchronization, only when the distance between them is close enough, i.e., under strong coupling strength[22,60]. The in-phase synchronization is observed between flagella on a single cell, with less variation in the beating frequency due to the internal connection[14]. The in-phase and out-of-phase modes in

the single cell occur at the regime of distinctly different beating frequencies, similar to the regimes of different flow rates in our system[14,30], including a phase slip with a noise, which is similar to the O–I transition regime[28].

In conclusion, in addition to the frequent encounter of the out-of-phase synchronization mode in droplet generation from two interfaces of immiscible fluids, we discover the regime of in-phase synchronization with phase locking in contrast to the non-synchronous regime from the weak coupling. The discovery of these overlooked regimes in the microfluidic T junction completes the state diagram of coupled oscillators accompanying small noises. Our experiments and theoretical model elucidate that the coupling depends on the distance between the interfaces which governs the forces acting on the interfaces, and also explain the transition between the in-phase and out-of-phase modes of droplet breakup. Therefore, the microfluidic system of oscillating interfaces described in this work serves as an instructive model for studying hydro-dynamic interactions leading to synchronization, with adjustable coupling strength using confinement and flow rates. For applica-tions, the microfluidic technology capable of inducing such stable synchronization or desynchronization of droplet generation pro-vides a new avenue for the valveless control of delivery and trig-gered reactions of materials in microdroplet formats[61,62].

## Methods

**Microchannel preparation**. The microfluidic channel was made by casting poly-dimethylsiloxane (PDMS; Sylgard 184 Silicone Elastomer Kit, Dow) onto a mold of microchannels, which was fabricated from SU-8 photoresist (MicroChem) on a silicon wafer by photolithography, and bonding it to another PDMS slab. The width of the main channel ($w_c$) was varied from 100 to 1500 μm. The width of the branch channel ($w_d$) for the water phase was fixed at 100 μm, and the height of all the channels was 40 μm.

**Materials and experiment setup**. Mineral oil with 2 wt.% Span80 (Sigma-Aldrich) as a surfactant was used as the continuous phase, and deionized (DI) water (18.2 MΩ cm) was used as the dispersed phase. The interfacial tension between the oil and water phases was measured to be $4.20 \pm 0.15$ mN m$^{-1}$. For the asymmetric droplet generation with different viscosities, we used a solution of 60 wt.% glycerol in DI water as one of the dispersed phases (dynamic viscosity: 10.5 cP at 21.5 °C). We used either syringe pumps (LEGATO® 100, KD Scientific, Inc.) at constant volumetric flow rates, or pressure pumps (Flow-EZ, Fluigent, and Elve-flow) at constant pressure, to flow the oil and water phases in all experiments, varying $P_o$ in the range of 200–1000 mbars, and $P_1$ and $P_2$ in the range of 165–855 mbars. The Reynolds number, Re $= \frac{\rho_o Q_o}{\mu_o h w_c L}$, where $\rho_o$ is the density of the con-tinuous phase and $L$ is set to be 100 μm, was maintained below 0.1 at all times. We observed the oscillation of the interface and droplet generation using an inverted microscope (DMi8, Leica) and recorded movies at 300–500 frames/s with a high-speed camera (Phantom VEO E310L, Vision Research) for about 20–30 s under each condition. The protrusion height and the droplet-breakup time, including the determination of the end of the filling stage for each interface, were analyzed from the recorded movies using custom macros of the ImageJ (NIH).

## Data availability
The data that support the findings of this study are available from the corresponding authors upon request.

## Code availability
The Mathematica code to generate numerical simulation results and datasets that support the findings of this study are available from the corresponding authors upon request.

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

## Acknowledgements

We thank Su Hyun Jung (UNIST) for helpful discussions. This work was supported by Basic Science Research Program through the National Research Foundation of Korea (NRF-2017R1A6A3A04006179). J.J. acknowledges support from 2019 Research Fund (1.190122.01) of UNIST (Ulsan National Institute of Science and Technology). H.A.S. thanks to the National Science Foundation (CMMI-1661672).

## Author contributions

E.U. and J.J. initiated this work. E.U., M.K., and H.K. performed the experiments. E.U., H.A.S., and J.J. set up the theoretical model and performed the numerical calculations. E.U., M.K., H.K., J.H.K. H.A.S., and J.J. analyzed the data and discussed the results. E.U., H.K. J.H.K., H.A.S., and J.J. wrote the paper.

## Competing interests

The authors declare no competing interests.
