## [Peer Review File · Nature Communications]

Reviewers' comments:

Reviewer #1 (Remarks to the Author):

The authors investigate synchronized droplet generation in a microfluidic device. Droplets are generated within the same channel at a double T-junction. The authors report synchronization of the droplet generation at the two T-junctions. The main result is that the author can switch the synchronization from in-phase synchronization to anti-phase synchronization by changing the ratio of flow rates. The authors develop a simple model that rationalizes the stability of in-phase synchronization and qualitatively describes the transition from in-phase to anti-phase synchrony.

The main result of the study is new (controlled transition from in-phase to out-of-phase synchrony), the experimental results are interesting and the analysis rigorous. I have reservation regarding the broader scope of the study. The synchronization in droplet generation has been reported in many studies, appropriately acknowledged and cited by the authors, including the anti-phase synchronization in a similar geometry (refs 44, 48) and hydrodynamic interactions in droplet microfluidics have been reported to support not only synchronization but also more complex repeatable logical operations (ref 37). The authors consider a particular geometry (droplet generation in a single channel and same flow rate assigned to the dispersed phase), which leads to (1) extremely strong interaction forces, due to the incompressibility condition and the strong geometric constraints of the single channel (indeed, the transition criterion from in-phase to out-of-phase synchrony is a geometrical constraint $b_{max} > 0.5$) (2) very low noise in the system. Therefore, while the contribution is very valuable, I do not think that the system has general and strong implications for other systems of synchronization in microfluidics or to the synchronization of cilia, which focuses on noisy oscillators coupled by weaker interactions (refs 22, 27).

I would also like to comment on transition criterion derived on page 11/12. While the experimental data for $w_c = 100$ microns qualitatively agrees with it, I do not think that the results for 300 and 400 microns do, and that the rescaling of the Capillary number leads to any collapse in fig 4. The criterion based on the geometric argument $b_{max} > 0.5$ seems to me to be valid for more constraint systems (smaller w_c), than for the wider channel. Plotting the criterion on fig. 2b and c, does not really show agreement.

Reviewer #2 (Remarks to the Author):

The authors perform experimental and theoretical studies of droplets formation in a microfluidic setup. While I appreciate the study as sound and potentially deserving publication, I think that the terminology used is completely misleading. In this situation, in my opinion, one cannot speak about synchronization, rather about two modes of operation - symmetric and asymmetric (in terms of synchronization theory these two modes would correspond to in-phase and anti-phase regimes). There is no evidence that two droplet-formation systems, operating completely separately, would produce periodic patterns with different periods, and their interaction leads to entrainment of these periods. Therefore I suggest resubmitting the manuscript, adopting a more appropriate terminology.

Reviewer #1

Comment #1: The main result of the study is new (controlled transition from in-phase to out-of-phase synchrony), the experimental results are interesting and the analysis rigorous. ... I have reservation regarding the broader scope of the study. The synchronization in droplet generation has been reported in many studies, ... I do not think that the system has general and strong implications for other systems of synchronization in microfluidics or to the synchronization of cilia, which focuses on noisy oscillators coupled by weaker interactions.

Response #1: We thank the reviewer for finding our results interesting and the contribution valuable. We improved our manuscript based on your comments as follows.

First of all, we want to emphasize that not only the transition from in-phase to out-of-phase synchronization is new, but also our report on the “in-phase” synchronization in the microfluidic droplet system is a significant first, as opposed to the “out-of-phase” synchronization reported in previous literatures (Refs. 46-51), where the regime of in-phase synchronization was overlooked.

In the revised manuscript, we emphasize the broad scope of this work and its general implication for other systems exhibiting hydrodynamic synchronization, throughout the introduction and discussion. We changed aspects of the wording from the previous manuscript. “Synchronization” was changed to “in-phase synchronization”, and “alternating” was changed to “out-of-phase synchronization”, and “uncoupled” to “nonsynchronous” regime. We also modified the title of the manuscript to “*Phase synchronization of fluid interfaces as hydrodynamically coupled oscillators.*”

In the previous manuscript, we mainly focused on reporting the observation of the in-phase synchronization in two-phase microfluidic system, which is reported for the first time in this work, but the characteristic of our system as a hydrodynamic coupled oscillator in a broader sense was not sufficiently illustrated. In the revised manuscript, we clarify that the microfluidic model system exhibits general regimes of two coupled oscillators, including the in-phase, out-of-phase, and transition between them, with the phase locking between the interfaces depending on the coupling strength, i.e., distance between the interfaces and flow rates. We also show the emergence of nonsynchronous regime where no phase locking is observed from the same microfluidic configuration, when interfaces are not coupled.

We further point out that the synchronization regime shown in our work exhibits non-zero phase differences of various ranges (Supplementary Fig. 6), that would occur in natural oscillators in a broad sense. The presence of all the regimes including the nonsynchronous and transition regime, indicates that our two-phase system is not simple two identical coupled oscillators (Ref. 10, 26) which display only two coupled states: perfect in-phase and anti-phase of 180-degree phase difference. Furthermore, we performed additional experiments assigning different flow rates to each branch, and studied the emergence of in-phase synchronization under variations of the flow, and m:n ratio generation of droplets as well. Therefore, the two-phase microfluidic model system can help study the mechanism of general hydrodynamic coupled oscillators of various synchronization regimes with phase delays. The control of such stable synchronization with adjustable phase delays in droplet generation provides a new avenue for the valveless control of delivery and triggered reactions of materials in microdroplet formats.

Changes #1

=> We added the description of “nonsynchronous regime” in Fig. 2a-c, and also the characteristic volume changes over time in Supplementary Fig. 2 to show there is no phase locking observed in this regime, even with the same flow rate and geometric configuration of the microchannel.

=> We added Supplementary Fig. 6 to show the variations in phase delay during phase locking in synchronization regimes.

=> Please refer to Response #2 for more changes in detail.

Comment #2: The authors consider a particular geometry ... which leads to (1) extremely strong interaction forces, ... (2) very low noise in the system.

Response #2: We thank the reviewer for pointing out these issues and help us improve the manuscript. In the revised manuscript, we clarify our exploration of *wide range of coupling strength* via changing the distance between two oscillating interfaces, i.e., the main channel width. This indeed results in various regimes from in-phase and out-of-phase synchronization to nonsynchronous state. This strategy of controlling the coupling strength via changing the distance between the oscillator is straightforward and reported in the renowned experiment of synchronization between two flagella on two different cells (Ref. 23). The anti-phase synchronization of the flagella results from the strong coupling, when the ratio of the distance d between the flagella to the length L , d/L is less than 1 ($d = 14.2 \mu\text{m}$), whereas the in-phase synchronization is observed between flagella within a single cell, that are internally linked with physical structure (Ref. 14). Our double T-junction enable us to explore all these synchronization regimes systematically with adjustment of parameters, and study the role of hydrodynamic interaction on synchronization, while excluding biological or chemical effects.

As with other system found in nature, the oscillating interfaces in our system also have variations in the frequency of generating droplets, although we set the flow rate to each branch to be the same. Specifically, in our microfluidic system, the droplet generation frequency when the interfaces are not coupled, had the coefficient of variation (CV) up to 13.5%. The CV in typical microfluidic T-channel operated with syringe pumps is reported to be about 10% (Ref. 52). This stochastic variation corresponds to other systems' noise of biological or chemical origins. The CV of the natural beating frequency of flagella on different cells that displayed the synchronization behavior (anti-phase) was around 10% (Ref. 23, 56), and many examples of the in-phase synchronization in flagella occurred within one cell, which are internally linked and thus the frequency distribution is expected to be narrower (Ref. 14, 28, 30). The noise contributes to the observed deviations in the phase difference from the two coupled states of identical coupled oscillators, i.e., the in-phase or anti-phase synchronization, and also the presence of the transition regime between the in-phase and out-of-phase.

Furthermore, we present additional experiments to investigate explicitly the effects of variation between two oscillators. Instead of sharing one flow source, we control the branches independently with two different pressure pumps, and measure how the in-phase synchronization is maintained during the change in the droplet volumes from the branches. As a result, we discover that the in-phase synchronization occurs with interfaces oscillating at lower noise level, but at larger distance compared to the out-of-phase synchronization. The out-of-phase synchronization can occur with broader variations in the branches but only at the closer distance of stronger coupling strength.

We can find implication of this study in other hydrodynamic coupling system as in the out-of-phase synchronization of flagella observed from different cells depending on the distance, by G.I. Taylor (Ref. 26) and D. R. Brumley (Ref. 23), and the in-phase synchronization occurring with flagella within one cell. (Ref. 14) Also, the transition between in-phase and out-of-phase resembles the phase slip observed from the flagella within one cell with some level of noises. (Ref. 28)

Changes #2:

=> We added the description of variations in droplet generation frequency, in Results section (Emergence of synchronized droplet generation from two interfaces; page 4-6), as follows.

“If there is no interaction between the two interfaces, each branch would generate droplets independently. ... the coefficient of variance of the droplet generation frequency, i.e., the standard deviation divided by the average, is 13.5% with a syringe pump,⁵² and 6.3% with a pressure pump; we conduct the statistical analysis with more than 500 droplets per each flow rate condition. Moreover, the difference in drop generation frequencies between the two branches is on average $7.27 \pm 3.71\%$ from 18 different flow rate conditions, indicating that the same Q_w does not guarantee the symmetric droplet generation mode. In this case, instead of synchronized droplet generation, we observe the decoupled droplet generation from two branches (Supplementary Fig. 2a), with two distinct frequency peaks (Supplementary Fig. 2b). The sharpness of the distinct peaks indicates that each branch generates droplets consistently but independently with no interference or phase locking between them. ... As in the out-of-phase synchronization mode, each branch in the in-phase mode exhibits a sharp frequency peak in the power spectrum, which overlaps exactly with the other within $\pm 0.01\%$ both in the in-phase and out-of-phase synchronization regimes due to phase locking (Supplementary Fig. 3c). Further reduction of Q_w will scramble this in-phase droplet generation and a state of a nonsynchronous regime appears.”

=> We added the results of the additional experiments of two independently controlled branches, in Discussion (1st and 2nd paragraphs; page 13-14):

“Our first observation of the in-phase synchronization of droplet breakup from two interfaces in a microchannel occurs with the same source of flow rates as to create nearly identical oscillators, with the intrinsic variance of about 10% in the droplet generation frequency from a T-channel.⁵² We investigate the effect of flow asymmetry on the appearance of in-phase synchronization by applying different pressures (P_1 on branch 1 and P_2 on branch 2) independently to each branch in $w_c = 300$ and $400 \mu\text{m}$ channels (Fig. 6a). ... After setting $P_1=P_2$ where the in-phase synchronization occurs (Fig. 6b, top) at every fixed pressure P_o to the oil phase, we only vary P_2 . The in-phase synchronization persists while the volume difference between two droplets reaches up to 75% (Fig. 6b, bottom, 6c and 6d). While satisfying the criterion for the in-phase synchronization, i.e., $0.8 < \alpha \leq 1$, the value of α tends to be larger when $P_2 > P_1$ from the branch 2, compared to $P_2 < P_1$ (Fig. 6d). ... The dependence of α on the volumetric configuration of droplets hints that the deviations in α from the ideal value $\alpha = 1$ in in-phase synchronization shown in Fig. 2e may stem from a slight symmetry breaking despite the symmetric design sharing the same flow source. As the asymmetry in the branches is increased with $|P_2-P_1| > 2$ mbar, the in-phase synchronization is perturbed, and eventually the O-I transition and the out-of-phase regime dominate.

For more practical use of our system with various fluids, we further assign different viscosities for the two dispersed phases as shown in Fig. 6e. ... where a 3:2 droplet generation ratio is observed. We find that this m:n droplet breakup mode, in fact, consists of both the out-of-phase and in-phase modes (indicated with blue arrows in Fig. 6f), manifesting the existence of hydrodynamic coupling between the interfaces.”

=> Variation in the coupling strength and the implication to other hydrodynamic synchronization phenomena is discussed in Discussion (3rd paragraphs; page 14-15):

“Under the influence of strong coupling strength tied to the distance ($b_{\text{max}}^* \geq 0.5$), the out-of-phase synchronization mode tolerates larger variations in the intrinsic frequencies than the in-phase mode, as similar to the previously reported work on droplet generation in parallel systems with slightly different flow rates.⁵¹ In order to achieve the in-phase synchronization, the coupling strength, which depends on the distance between the interfaces, has to be decreased ($b_{\text{max}}^* < 0.5$) and the frequencies of the two oscillating interfaces have to be similar, in the range of small variations. Our findings resemble a series of observations in hydrodynamically coupled oscillators,¹⁸ including flagella on cells,²⁴ having the coefficient of variation in their intrinsic frequencies as much as 10%.⁵⁶ Two flagella on two separate cells dominantly display the out-of-phase synchronization mode, only when the distance between them is as close as the length of a flagellum ($\sim 10 \mu\text{m}$).²³ The occurrence of in-phase synchronization mode was observed between flagella on a single cell with less variation, which is strongly related to their internal connection with fibers.¹⁴ Other observations include the in-phase and out-of-phase modes occurring at distinctly different beating frequency regime, similar to the different regime of flow rates in our system,³⁰ and a phase slip with a noise, which is similar to the O-I transition regime.²⁸”

Comment #3: ... to comment on transition criterion derived on page 11/12. ... that the rescaling of the Capillary number leads to any collapse in fig 4.

Response #3: In the revised manuscript, we propose a modified criterion to better describe all the experimental data from different channel widths. We agree that some data points, especially ones from the channels of larger widths at a smaller Capillary number show deviations from the proposed criterion line. In addition to providing possible explanations for the discrepancy in the main text following Fig. 4, we adopt a new set of dimensionless parameters, as shown in Fig. 4 with the y-axis in log scale, considering that the rate of protrusion of the interface, $db(t)/dt$ is not only related to Q_w , but also to the channel dimensions, w_i and w_c , for more accurate description. Moreover, to supplement the comparison between the experimental data and the criterion line, we include the data points of the transition regime in the plot, which are scattered around the criterion line. Considering the intrinsic variations in the experimental system, and bold assumptions in the model, we believe that this criterion can serve as a starting point to discuss the parameters affecting the transition from the in-phase to out-of-phase synchronization.

Changes #3:

=> In Result section (Transition from the in-phase to the out-of-phase state; page 11-12) and Fig. 4:

“For the left-hand side of equation (3), we can approximate $db(t)/dt \propto Q_w/(w_i+w_c) \cdot h$, because the protrusion rate of dispersed phase is related with flow rate Q_w , and the channel dimensions, i.e., the channel width w_c and branch width w_d (Supplementary Fig. 11). T_{neck} is proportional to $w_d w_c h / Q_o$, since the neck of the dispersed phase shrinks faster as the continuous phase flows faster and the branch width w_d gets narrower (Supplementary Fig. 10).^{53,54} ... After replacing T_{fill} with the solution of the force-balance equation, the equation of the transition boundary is written in terms of $Ca = \frac{\mu_o Q_o}{\gamma w_c h}$ as

$$\frac{Q_w/Q_o}{1+\Lambda} = C \cdot \frac{6}{11 + \sqrt{1 + \frac{48h}{w_c Ca}}} \quad (4)$$

where $\Lambda = w_c/w_d$ is the width ratio and C is a coefficient of proportionality. ... We plot the experimental data from Fig. 2a-c with the effective dimensionless numbers, i.e., $Ca \frac{w_c}{h}$ and $\frac{Q_w/Q_o}{1+\Lambda}$, and compare it with the trend of Equation (4) (Fig. 4), choosing $C = 0.12$ to best describe the collapsed data. ... Because the actual O-I transition occurs between $0.4 < b_{\text{max}}^* \leq 0.5$, the transition boundary in the experimental state diagrams is rather broad. Moreover, in this prediction of the transition boundary, we assume T_{neck} is inversely proportional to Q_o , since there is no accurate theory available to predict T_{neck} . However, as observed from experiments, the variation in T_{neck} gets larger as Q_o is reduced, which may result in a wider transition regime towards lower Ca . We also expect that different dynamics of droplet breakup may be involved for larger w_c . As seen from the difference in the limit of b_{max}^* in each confinement, e.g. $w_c = 400 \mu\text{m}$ and $1500 \mu\text{m}$, b_{max}^* never exceeds 0.5 and 0.2, respectively, regardless of flow rates, which cannot be explained with the current model. Since b_{max}^* is an important parameter to determine the transition from the alternating to synchronized states, the mechanism of determining b_{max}^* in each confinement should be investigated further.

„

Reviewer #2

Comment: While I appreciate the study as sound and potentially deserving publication, I think that the terminology used is completely misleading. In this situation, in my opinion, one cannot speak about synchronization, rather about two modes of operation – symmetric and asymmetric. ... There is no evidence that two droplet-formation systems, operating completely separately, would produce periodic patterns with different periods, and their interaction leads to entrainment of these periods.

Response: We thank the reviewer for finding our study sound and potentially deserving publication. The revised manuscript addresses the reviewer's concern, emphasizing the *phase locking* between two oscillating interfaces. We show the experimental evidence that when two branches are not coupled, each can generate droplets periodically but independently in different frequencies with variations (Supplementary Fig. 2). When the interfaces are coupled within the closer distance affecting each other, the frequencies coincide, and in-phase and out-of-phase synchronization of fixed phase delay α (Supplementary Fig. 6), or transition regime emerges depending of the coupling strength (Supplementary Fig. 3). Please note that in the revised manuscript, we changed aspects of the wording from the previous manuscript for clarification. "Synchronization" was changed to "in-phase synchronization", and "alternating" was changed to "out-of-phase synchronization", and "uncoupled" to "nonsynchronous" regime. We also modified the title of the manuscript to "Phase synchronization of fluid interfaces as hydrodynamically coupled oscillators."

Furthermore, we performed additional experiments controlling the flow rates in each branch independently with two different pressure pumps, instead of sharing one flow source, to investigate explicitly the effects of variations between two oscillators on the emergence of synchronization. Together with data observed with the branches controlled with the same source, and with independently controlled sources, we observe the phase locking when interfaces are coupled. The in-phase synchronization occurs with interfaces oscillating at lower noise level, but at larger distance compared to the out-of-phase synchronization. The out-of-phase synchronization can occur with broader variations in the branches but only at the closer distance of strong coupling strength.

We further point out that the synchronization regime shown in our work exhibits non-zero phase differences of various ranges (Supplementary Fig. 6), that would occur in natural oscillators in a broad sense. The presence of all the regimes including the nonsynchronous and transition regime, indicates that our two-phase system is not simple two identical coupled oscillators (Ref. 10, 26) which display only two coupled states: perfect in-phase and anti-phase of 180-degree phase difference. Therefore, the two-phase microfluidic model system in this work can help study the mechanism of general hydrodynamic coupled oscillators of various synchronization regimes with phase delays.

Changes:

=> We added the description of "nonsynchronous regime" in Fig. 2a-c, and also the characteristic volume changes over time in Supplementary Fig. 2 to show there is no phase locking observed in this regime, even with the same flow rate and geometric configuration of the microchannel.

=> We added Supplementary Fig. 6 to show the variations in phase delay during phase locking in synchronization regimes.

=> We added the description of variations in droplet generation frequency, in Results section (Emergence of synchronized droplet generation from two interfaces; page 4-6), as follows.

"If there is no interaction between the two interfaces, each branch would generate droplets independently. ... the coefficient of variance of the droplet generation frequency, i.e., the standard deviation divided by the average, is 13.5% with a syringe pump,³² and 6.3% with a pressure pump; we conduct the statistical analysis with more than 500 droplets per each flow rate condition. Moreover, the difference in drop generation frequencies between the two branches is on average $7.27 \pm 3.71\%$ from 18 different flow rate conditions, indicating that the same Q_w does not guarantee the symmetric droplet generation mode. In this case, instead of synchronized droplet generation, we observe the decoupled droplet generation from two branches (Supplementary Fig. 2a), with two distinct frequency peaks (Supplementary Fig. 2b). The sharpness of the distinct peaks indicates that each branch generates droplets consistently but independently with no interference or phase locking between them. ... As in the out-of-phase

synchronization mode, each branch in the in-phase mode exhibits a sharp frequency peak in the power spectrum, which overlaps exactly with the other within $\pm 0.01\%$ both in the in-phase and out-of-phase synchronization regimes due to phase locking (Supplementary Fig. 3c). Further reduction of Q_w will scramble this in-phase droplet generation and a state of a nonsynchronous regime appears.”

=> We added the results of the additional experiments of two independently controlled branches, in Discussion (1st and 2nd paragraphs; page 13-14):

“Our first observation of the in-phase synchronization of droplet breakup from two interfaces in a microchannel occurs with the same source of flow rates as to create nearly identical oscillators, with the intrinsic variance of about 10% in the droplet generation frequency from a T-channel.⁵² We investigate the effect of flow asymmetry on the appearance of in-phase synchronization by applying different pressures (P_1 on branch 1 and P_2 on branch 2) independently to each branch in $w_c = 300$ and $400 \mu\text{m}$ channels (Fig. 6a). ... After setting $P_1 = P_2$ where the in-phase synchronization occurs (Fig. 6b, top) at every fixed pressure P_o to the oil phase, we only vary P_2 . The in-phase synchronization persists while the volume difference between two droplets reaches up to 75% (Fig. 6b, bottom, 6c and 6d). While satisfying the criterion for the in-phase synchronization, i.e., $0.8 < \alpha \leq 1$, the value of α tends to be larger when $P_2 > P_1$ from the branch 2, compared to $P_2 < P_1$ (Fig. 6d). ... The dependence of α on the volumetric configuration of droplets hints that the deviations in α from the ideal value $\alpha = 1$ in in-phase synchronization shown in Fig. 2e may stem from a slight symmetry breaking despite the symmetric design sharing the same flow source. As the asymmetry in the branches is increased with $|P_2 - P_1| > 2$ mbar, the in-phase synchronization is perturbed, and eventually the O-I transition and the out-of-phase regime dominate.

For more practical use of our system with various fluids, we further assign different viscosities for the two dispersed phases as shown in Fig. 6e. ... where a 3:2 droplet generation ratio is observed. We find that this m:n droplet breakup mode, in fact, consists of both the out-of-phase and in-phase modes (indicated with blue arrows in Fig. 6f), manifesting the existence of hydrodynamic coupling between the interfaces.”

=> Variation in the coupling strength and the implication to other hydrodynamic synchronization phenomena is discussed in Discussion (3rd paragraphs; page 14-15):

“Under the influence of strong coupling strength tied to the distance ($b_{\text{max}}^* \geq 0.5$), the out-of-phase synchronization mode tolerates larger variations in the intrinsic frequencies than the in-phase mode, as similar to the previously reported work on droplet generation in parallel systems with slightly different flow rates.⁵¹ In order to achieve the in-phase synchronization, the coupling strength, which depends on the distance between the interfaces, has to be decreased ($b_{\text{max}}^* < 0.5$) and the frequencies of the two oscillating interfaces have to be similar, in the range of small variations. Our findings resemble a series of observations in hydrodynamically coupled oscillators,¹⁸ including flagella on cells,²⁴ having the coefficient of variation in their intrinsic frequencies as much as 10%.⁵⁶ Two flagella on two separate cells dominantly display the out-of-phase synchronization mode, only when the distance between them is as close as the length of a flagellum ($\sim 10 \mu\text{m}$).²³ The occurrence of in-phase synchronization mode was observed between flagella on a single cell with less variation, which is strongly related to their internal connection with fibers.¹⁴ Other observations include the in-phase and out-of-phase modes occurring at distinctly different beating frequency regime, similar to the different regime of flow rates in our system,³⁰ and a phase slip with a noise, which is similar to the O-I transition regime.²⁸

REVIEWERS' COMMENTS:

Reviewer #1 (Remarks to the Author):

I would like to thank the authors for their response, addressing the points raised in the review and for the clarifications brought to the manuscript. I continue to have reservations regarding the broader scope of the study. The authors stress the implications of this microfluidic system for flagellar synchronizations. I do not think that the results presented in the manuscript have implication or provide new insight into flagellar synchronization. The authors highlight some similarities, which is, to a certain extend, expected since both system synchronize, however similarities and implications are not the same.

Reviewer #2 (Remarks to the Author):

The authors responded to my remarks properly and I now recommend publication of the manuscript.

Reviewer #3 (Remarks to the Author):

The report here is written in the context of an appeal.

The experimental observations are neat, the data is organized in a rigorous manner, the statements are well supported by the numerous measurements performed by the authors, and the theory does not leave much doubt about its validity. It is true, as pointed out by a referee, that much work has already been performed on T-junctions (Garstecki,..). Less is known on T-junctions or V-junctions with a double droplet emission. Still, regimes with alternated phase-locked droplets, similar to the out-of-phase regime of the paper, were reported (an example is Herminghaus's work). T-junctions in shallow geometries with a single emitter is much less known, although, recently, a paper provided a description (Leshansky et al, 2019). So clearly, the situation considered by the authors, a double emitter in shallow microchannels, is new. In their work, the authors observed a regime where droplets move side by side. This regime was quite unexpected and, in my opinion, it was a tour-de-force to provide a theoretical description, that figures out the essential mechanisms at work. In my knowledge, this contribution is unique, all preceding work restricting themselves to phenomenological observations or abstract modeling. Whether the scope of a paper is broad, a question raised by a reviewer, is often delicate to discuss. In my opinion, the situation considered by the authors is generic, their work will represent a reference for the growing field of microfluidics, along with a source of inspiration, conceptual and technical, for modeling more complex phenomena arising in the living world, such as cilia movements. The second referee raises more technical questions about synchronization. Two independent emitters will never give rise to the phase locking phenomena observed in the paper, we need nonlinear interactions between them, which the paper precisely succeeds to figure out. To conclude, I support its publication in Nature Communications.

Reviewer #1: I would like to thank the authors for their response, addressing the points raised in the review and for the clarifications brought to the manuscript. I continue to have reservations regarding the broader scope of the study. The authors stress the implications of this microfluidic system for flagellar synchronizations. I do not think that the results presented in the manuscript have implication or provide new insight into flagellar synchronization. The authors highlight some similarities, which is, to a certain extend, expected since both system synchronize, however similarities and implications are not the same.

Response: We thank the reviewer for the valuable comments to improve the manuscript.

In this work, we show that our system can serve as an experimental model system of hydrodynamically coupled oscillators, and the theoretical explanations on the mechanism can deepen our understanding of the hydrodynamic interactions in the synchronized behaviors. We believe that our work has made a unique contribution to the field of microfluidics, as well as to the growing fields of applications in fluid mechanics and biomedical engineering. Controlling the droplet generation with adjustable phase delays, along with our additional experiments on the independently controlled fluid interfaces of different characteristics, suggests possible future applications of the system, such as valveless control of microdroplets for spatiotemporal delivery and triggered reactions of materials.

As Reviewer#1 mentioned, we find similarities in the synchronization of flagellar movement and our fluid-fluid interfaces, as both phenomena encompass the in-phase and out-of-phase synchronization, and transitions with the influence of noises, as described in Discussion (the 3rd paragraph). However, we do not intend to implicate our work can explain the synchronization phenomena of biological cells. In order to clarify the scope of our work, we changed Abstract as follows:

From “Thus, in this study we provide a model system to deepen our understanding of the role of hydrodynamic interactions for synchronized motions in fluids, such as phase synchronization of flagella, and may benefit the design of future microfluidic devices, allowing spatiotemporal control of microdroplet generation without additional integration of control elements.”

to “Hydrodynamic interactions play a role in synchronized motions of coupled oscillators in fluids, and understanding the mechanism will facilitate development of applications in fluid mechanics. For example, synchronization phenomenon in two-phase flow will benefit the design of future microfluidic devices, allowing spatiotemporal control of microdroplet generation without additional integration of control elements.”

Reviewer #2: The authors responded to my remarks properly and I now recommend publication of the manuscript.

Response: We thank the reviewer for the valuable comments to improve the paper and recommending our manuscript for publication.

Reviewer #3: The report here is written in the context of an appeal.

The experimental observations are neat, the data is organized in a rigorous manner, the statements are well supported by the numerous measurements performed by the authors, and the theory does not leave much doubt about its validity. It is true, as pointed out by a referee, that much work has already been performed on T-junctions (Garstecki,..). Less is known on T-junctions or V-junctions with a double droplet emission. Still, regimes with alternated phase-locked droplets, similar to the out-of-phase regime of the paper, were reported (an example is Herminghaus's work). T-junctions in shallow geometries with a single emitter is much less known, although, recently, a paper provided a description (Leshansky et al, 2019). So clearly, the situation considered by the authors, a double emitter in shallow microchannels, is new. In their work, the authors observed a regime where droplets move side by side. This regime was quite unexpected and, in my opinion, it was a tour-de-force to provide a theoretical description, that figures out the essential mechanisms at work. In my knowledge, this contribution is unique, all preceding work restricting themselves to phenomenological observations or abstract modeling. Whether the scope of a paper is broad, a question raised by a reviewer, is often delicate to discuss. In my opinion, the situation considered by the authors is generic, their work will represent a reference for the growing field of microfluidics, along with a source of inspiration, conceptual and technical, for modeling more complex phenomena arising in the living world, such as cilia movements. The second referee raises more technical questions about synchronization. Two independent emitters will never give rise to the phase locking phenomena observed in the paper, we need nonlinear interactions between them, which the paper precisely succeeds to figure out.

To conclude, I support its publication in Nature Communications.

Response: We thank the reviewer for providing valuable comments and consider our work unique and generic. To acknowledge previous works using microfluidic T-channels in Introduction more thoroughly, we added the following references on the study of droplet generation in T-channels and on the alternating droplet generation, in Introduction (the 3rd paragraph) as follows.

“The mechanism of droplet generation in a T-channel of confined geometry has been studied frequently.⁴⁰⁻⁴⁵ ... In a microfluidic configuration where two interfaces interact,⁵¹ including the so-called double T-junction geometry,^{52,53} the implementation of one-by-one generation of droplets has triggered various applications that exploit the controlled delivery of droplets of two different compositions.^{53-57”}

- 40 Chakraborty, I. *et al.* Droplet generation at Hele-Shaw microfluidic T-junction. *Phys. Fluids* **31**, 022010, (2019).
- 45 De Menech, M., Garstecki, P., Jousse, F. & Stone, H. A. Transition from squeezing to dripping in a microfluidic T-shaped junction. *J. Fluid Mech.* **595**, 141-161 (2008).
- 51 Chokkalingam, V., Herminghaus, S. & Seemann, R. Self-synchronizing pairwise production of monodisperse droplets by microfluidic step emulsification. *Appl. Phys. Lett.* **93**, 254101 (2008).
- 54 Chokkalingam, V. *et al.* Optimized droplet-based microfluidics scheme for sol-gel reactions. *Lab Chip* **10**, 1700-1705 (2010).